# Circadian Regulation of Retinal Pigment Epithelium Function

**DOI:** 10.3390/ijms23052699

**Published:** 2022-02-28

**Authors:** Kenkichi Baba, Varunika Goyal, Gianluca Tosini

**Affiliations:** Department of Pharmacology & Toxicology and Neuroscience Institute, Morehouse School of Medicine, Atlanta, GA 30310-1495, USA; bkenkichi@msm.edu (K.B.); vgoyal@msm.edu (V.G.)

**Keywords:** circadian, melatonin, dopamine, RPE, retina, phagocytosis

## Abstract

The retinal pigment epithelium (RPE) is a single layer of cells located between the choriocapillaris vessels and the light-sensitive photoreceptors in the outer retina. The RPE performs physiological processes necessary for the maintenance and support of photoreceptors and visual function. Among the many functions performed by the RPE, the timing of the peak in phagocytic activity by the RPE of the photoreceptor outer segments that occurs 1–2 h. after the onset of light has captured the interest of many investigators and has thus been intensively studied. Several studies have shown that this burst in phagocytic activity by the RPE is under circadian control and is present in nocturnal and diurnal species and rod and cone photoreceptors. Previous investigations have demonstrated that a functional circadian clock exists within multiple retinal cell types and RPE cells. However, the anatomical location of the circadian controlling this activity is not clear. Experimental evidence indicates that the circadian clock, melatonin, dopamine, and integrin signaling play a key role in controlling this rhythm. A series of very recent studies report that the circadian clock in the RPE controls the daily peak in phagocytic activity. However, the loss of the burst in phagocytic activity after light onset does not result in photoreceptor or RPE deterioration during aging. In the current review, we summarized the current knowledge on the mechanism controlling this phenomenon and the physiological role of this peak.

## 1. Introduction

Circadian rhythms are important features of living organisms. These rhythms are endogenously generated by molecular clocks (i.e., the circadian clocks) that have a period close to 24 h. This circadian oscillation is generated by a transcriptional translational feedback loop that involves several “clock” genes and their products (Figure 1). Numerous studies have also shown that circadian clocks are present in several ocular tissues (e.g., retina, cornea, retinal pigment epithelium, etc.) where they control critical physiological functions (see [1] for a recent review and references within). Additional studies have also reported that dysfunction of the circadian clock and its outputs (i.e., melatonin, dopamine, etc.) in the eye may adversely affect these tissues [2,3,4,5,6,7,8,9,10].

The retinal pigment epithelium (RPE) is a single layer of cells located between the choriocapillaris vessels and the light-sensitive photoreceptors in the outer retina. The RPE performs physiological processes necessary for the maintenance and support of photoreceptors and visual function. Among the several functions of the RPE, the role in the continuous renewal of the light-sensitive outer segment of photoreceptors is critical for the photoreceptor health [11]. Photoreceptors synthesize new outer segment components and form new outer segment disks. A process commonly termed disk shedding compensates for this addition during which RPE cells, in collaboration with photoreceptors, remove the most distal tip of rod and cone outer segments (POS) [12,13]. The POS sheds and then is promptly engulfed by the RPE and degraded with this tissue. Previous studies show that the lack of phagocytosis by the RPE leads to the accumulation of POS in the subretinal space and consequently to photoreceptor degeneration [14]. The present review aimed to summarize the current knowledge about the circadian regulation of RPE function and how disruption of the circadian function may lead to many pathologies.

## 2. Regulation of RPE Function In Vivo

The process of photoreceptor outer segment shedding and phagocytosis follows a diurnal rhythm characterized by a burst/peak in the level of phagocytosis 1–2 h after the onset of light [15]. This rhythm persists under constant darkness [16,17] or light [18], after the optic nerve has been severed [19], and once the master circadian clock located in the brain has been ablated [20]. Because of this experimental evidence, it was proposed that the rhythms in photoreceptor disc shedding, and phagocytosis were under the control of circadian clocks located within the eye [16,19,20]. Indeed, later studies demonstrated the presence of circadian clocks in the retina [21], within the photoreceptors [22,23], the inner retina [23,24], and within the RPE [25,26]. However, it is still unclear whether the rhythm in phagocytosis by the RPE is controlled by the circadian clock located in the retina, in the RPE, or both. It is also worth noting that although most of the studies on RPE phagocytic activity have focused on the process controlling phagocytosis in the rod photoreceptors, a circadian process has also been described for the cone photoreceptors [27,28]. Since the burst in phagocytic activity is present in many vertebrate species (including humans) and other experimental conditions, it was hypothesized that the presence and timing of the peak in the phagocytic activity must play an important role in the health of the photoreceptors and the RPE [1]. However, it is important to mention that although several studies have indicated that phagocytic activity by the RPE play and important role in maintain a healthy retina in humans (Table 1) no study so far has indicated that dysfunction the phagocytic rhythm in humans is a cofactor in the development of human retinal pathologies.

Hence, in the last twenty years, many laboratories investigated the molecular mechanisms controlling the burst in the phagocytic activity and then have produced a new animal model to prove that the presence of this peak was indeed important for the overall health of the RPE and the photoreceptors. These studies reported a lack the activation of key agents of phagocytic signaling, including focal adhesion kinase (FAK) [29,30,31], MerTK [29,30], and Rac1 GTPase, a potent F-actin regulator [31,32], to be the underlying cause of the disturbed phagocytosis process by the RPE. In 2012, Ruggiero et al. [33] also reported that the rods expose a conserved phosphatidylserine domain at their distal tip, which serves as an “eat me” signal at the light onset. Interestingly, this phosphatidylserine exposure is not rhythmic in mice lacking the diurnal rhythm of RPE phagocytosis due to a lack of *α*v*β*5 integrin receptor or its ligand Milk Fat Globule-EGF factor 8 protein (MFG-E8), thus suggesting that the key signal initiating the burst of phagocytic activity is from the RPE. Other studies have also implicated dopamine and melatonin signaling in this process since Dopamine 2 Receptors (D_2_R) knock-out (KO) mice do not show a daily peak in phagocytic activity [31], and melatonin receptor KO mice show a disruption in the timing of the peak [34]. However, it is important to mention that, in all these studies, the total daily phagocytic activity was not different from the wild-type mice. Other investigations have also revealed that mice lacking myosin VIIa, Annexin A2, and lysosomal protein melanoregulin showed a normal phagocytic peak but delay in trafficking from the apical to the basal region of the RPE [35,36].

To gather further insights into the molecular mechanism and signaling pathways controlling the circadian rhythm of the phagocytic activity in the RPE, several studies have conducted transcriptomic profiling of this tissue at various time points before, during, and after the phagocytosis process. In one such study in 2013, Mustafi et al. [37] found that several genes (*Dgki*, *Itpr1*, *Pik3r1*, *Lamp2* and *Pla2g2*) involved in polyphosphoinositide signaling are upregulated in mice RPE 1.5 h. after the onset of light. Along with that, the promoter region of these genes also contains CLOCK: BMAL1 and ROR binding motifs, thus reinforcing the notion that the circadian clock in the RPE is modulating the phosphoinositide signaling. An additional study using RNA-sequencing and pathway analysis reported that approximately 20% of the RPE transcriptome is under circadian regulation [38]. This study also showed that one hour after the subjective light onset, pathways associated with RPE phagocytosis, including integrin signaling, cAMP signaling, focal adhesion, epithelial adherence junction signaling, mitochondrial phosphorylation, and protein phosphorylation, are upregulated in the RPE [38]. The circadian regulation of the RPE transcriptome also involves the regulation of metabolic pathways since the transcription of several genes involved in ATP (Adenosine triphosphate) production, fat metabolism, and other metabolic pathways are under circadian control [39]. In the middle of the night, transcripts involved in the mitochondrial ETC (electron transport chain), TCA (tricarboxylic acid) cycle, glycolysis, and glycogen metabolism were higher, suggesting a greater need for energy production at this time point in the RPE. Additionally, an up-regulation of genes implicated in the glycerophospholipid metabolism pathway was also observed at this time. Glycerophospholipids are the building blocks of membrane lipids. An upregulation in the metabolism of glycerophospholipids at night might suggest that they could be transported to photoreceptor cells to form new outer space segments [40] as soon as the exposed outer segments are shed. Conversely, the fatty acid degradation pathway was also upregulated during the day as the RPE digests the lipid layer of the ingested POS.

Although several studies have investigated the effects of clock genes removal in the retina, only two investigations have examined the effect of clock gene removal on the circadian regulation of phagocytic activity. In the first study, *Per1/Per2* global KO blunted the peak in phagocytic activity, and 57 genes involved in photoreceptor phagocytosis were downregulated in the RPE [41]. In the second study, DeVera et al. [42] developed an RPE-specific *Bmal1* KO mouse model. They demonstrated that the circadian clock in the RPE controls the daily diurnal peak in phagocytosis of POS since, in retina specific *Bmal1* KO, the daily rhythm in phagocytic activity was not affected by the removal of this gene. In contrast, the removal of *Bmal1* from the RPE abolished the daily rhythm [42].

**Table 1 ijms-23-02699-t001:** List of protein that have been associated with the daily rhythm in phagocytic activity and their involvement in human retinal diseases.

	Function in RPE	Animal Model	Human Retinal Disease
**MerTK receptor**	Outer segment binding & internalization	RCS rat [43,44] Mer^kd^ mouse [45]	Retinitis pigmentosa, rod-cone dystrophy [46,47,48,49,50]
**Gas6, Protein S**	MerTK ligands	Gas6 double KO and ProS1 [51]	Diabetic Retinopathy and macular edema [52]
**ανβ** **5 integrin receptor**	Outer segment binding, Control the diurnal rhythm in peak of phagocytosis	β5^−/−^ mouse [29]	unknown
**MFG-E8**	ανβ5 integrin ligand Control the diurnal rhythm in peak of phagocytosis	MFG-E8^−/−^ mouse [30]	unknown
**Dopamine receptor 2**	Controls the rhythm in RPE circadian clocks, light adaptation, peak of phagocytosis after light onset	D_2_R KO mouse [31]	unkown
**Melatonin receptor 1 and 2**	Control the timing of the peak of phagocytosis	MT_1_ & MT_2_ KO mouse [34]	
**RPE specific Bmal1 KO**	Control the diurnal rhythm in peak of phagocytosis	RPE^cre^; *Bmal1*^fl/fl^ [42]	unknown
***Per1/Per2* global KO**	Controls the amplitude of the peak of phagocytosis	*Per1*^−/−^*Per2^Brdm1^* [41]	unknown

## 3. Regulation of RPE Function In Vitro

In the last few years, a few methods have been introduced to culture primary RPE cells from many species, including humans [53,54]. Some procedures include an RPE sheet peeled from the choroidal cup for large animals [55,56]. However, isolating RPE sheets from small animals, such as the mouse, is still a challenge [25,57]. As an alternative to primary RPE culture, a few immortalized RPE cell lines have been produced. The most used RPE cell line is ARPE-19, which was established from cells isolated from the enucleated globes of a 19-year-old male donor [58]. In addition, the RPE-J cell line is derived from a 7-day-old Long-Evans rat and is commonly used for non-human RPE studies [59]. While these cell lines show similar phenotypical features as native RPE cells, the cell lines have some physiological and morphological differences from native cells, such as a lack of pigmentation and turnover [60,61,62].

The first circadian clock oscillation in in vitro RPE cells was observed in human RPE cell lines (*h*RPE). In this study, *h*RPE cells were first synchronized with forskolin, and then the cells were collected every 6 h. The authors reported *Per1* and *Per2* mRNA levels were rhythmically transcribed for at least three circadian cycles [63]. Then, Yoshikawa et al. [64], using a *Bmal1*-luciferase bioluminescence reporter system, demonstrated a circadian rhythm in *Bmal1-luciferase* bioluminescence for 7 days in *h*RPE cells. More recent findings also revealed that several genes involved in RPE phagocytic activity are also expressed in a circadian manner in the ARPE19 cell line [65]. This research team also reported that the culture methodology (dispersed vs. monolayer) might affect clock gene rhythmic expression amplitude in ARPE19 cells.

The recent introduction of bioluminescence reporter technologies and the generation of transgenic mice in which bioluminescence can be recorded from tissue explants have led to significant advances in understanding the mammalian circadian system [66]. Using these newly generated transgenic mice, circadian rhythms have been observed in the retina, photoreceptor, cornea, iris-ciliary body, and RPE explants collected from PERIOD2::LUCIFERASE (PER2::LUC) mice eyes [23,24,25,26,67,68,69]. Using this mouse model, in 2010, Baba et al. showed that although a clear bioluminescence rhythm can be observed from isolated RPE cells, this rhythm has a larger amplitude and the RPE survive better when cultured together with choroid [25]. The phase and period of mice RPE PER2::LUC rhythm is slightly different from the retinal PER2::LUC rhythm (RPE vs. Retina Phases: ZT 16.5 h vs. ZT 12.4 h, light onset as ZT 0/dark onset as ZT 12; Periods: 23.9 h vs. 24.3 h). Moreover, this circadian rhythm can persist for more than fifty days with weekly medium exchanges [25]. Finally, it is worth mentioning that the age of the mice and the time of the day at which the tissue is explanted and prepared for the culture preparation may affect the circadian parameters, and thus it should be carefully considered [7,70].

## 4. Entrainment of the RPE Circadian Clock

A few studies have reported that light can entrain the circadian rhythms of the isolated retina and cornea from mice [24,71], but light does not appear to entrain the circadian rhythm of bioluminescence in the mouse RPE [25,72]. In the eye, melatonin and dopamine play an important role in the regulation and entrainment of circadian rhythms [1], and, as previously mentioned, melatonin signaling is involved in the regulation of the timing and amplitude of the morning phagocytic peak [31,33,73]. Several lines of evidence also suggest that dopamine and its receptors are involved in regulating rhythmic RPE function. For example, the inhibition of dopamine synthesis during the early part of the light phase induced a significant reduction of disk shedding and phagocytosis [74], and mice whose dopaminergic neurons have been destroyed by 1-methyl-4-phenyl-1,2,3,6-tetrahydropyridine (MPTP) accumulate many residual bodies in the RPE [75].

This experimental evidence has led to the notion that these two neurohormones may be responsible for the entrainment of the RPE circadian clock. Indeed, Baba et al. [26] demonstrated that dopamine, but not melatonin, entrained the circadian rhythms in PER2::LUC bioluminescence in cultured RPE. The administration of exogenous dopamine (100 mM) entrains the circadian rhythm in the RPE in a phase-dependent manner. Moreover, it delays the RPE PER2::LUC rhythm during late night to early morning and phase advances PER2::LUC rhythm during the day [26]. Such an effect appears to be exclusively mediated by D_2_R signaling [26]. The mechanisms by which D_2_R affects entraining the circadian clock in the RPE are not yet understood since D_2_R signaling is negatively coupled to an adenylyl cyclase, which leads to reduced cAMP levels. Since previous studies have shown that increases in cAMP levels are indeed involved in the clock resetting mechanism by acting on cAMP responsive elements in the *Period* gene promoter [76,77,78] it must be concluded that it is unlikely that D_2_R receptor activation, which reduces cAMP levels, induces the phase shifts via the cAMP cascade. Moreover, it is worth mentioning that a previous study indicated that D_2_R signaling induces the phosphorylation of the cAMP Response Element Binding Protein (CREB) by activating Ca^2+^/calmodulin-dependent protein kinase [79] and the ERK1/2 signaling pathway [80,81,82]. ERK1/2 signaling has also been implicated in the mechanism of the entrainment of the master circadian clock [83,84,85,86]. Further studies also show that the 90RSK protein plays a key role in activating the ERK-CREB pathway [87,88]. Thus, we propose that the mechanism responsible for the entrainment of the RPE via D_2_R involves the ERK1/2/ pathway, as described in Figure 2.

An alternative mechanism for the entrainment of the circadian clock in the RPE was also proposed. According to Ikarashi et al. [72], cytosolic Ca^2+^ level shows circadian variation in RPE cells, and this circadian variation in Ca^2+^ levels is abolished in *Bmal1* dominant-negative RPE cells. This study also showed that muscarinic receptor subtype, M3, is expressed in *h*RPE cells and the acetylcholine/carbachol-induced Ca^2^ elevation seems to be under circadian control. Finally, they reported that the administration of carbachol successfully phase-shifted *Bmal1-luc* rhythms in *h*RPE cells in a phase-dependent manner [72]. Additionally, histamine, another element known to increase cytosolic Ca^2+^ levels, phase-shifts *Bmal1-luc* rhythms in *h*RPE cells in a phase-dependent manner very similar to the carbachol-induced phase-shift [72]. This effect was abolished when histamine was treated with H1 receptor antagonists. In contrast, treatment with an H2 receptor agonist showed only a small effect. Thus, H1 receptor signaling is probably more dominant in mediating RPE circadian phase-shifts in *h*RPE cells [89]. Interestingly, another research group showed that three hours of a co-incubation with POS induced lysosomal marker, LAMP1, expression in cultured ARPE19 cells in a time-dependent manner [41]. Hence, this study indicates that the circadian clock regulates the phagocytic activity of the RPE, and photoreceptor outer binding availability and perhaps the phagocytic activity itself may provide feedback to the circadian clock in the RPE. Thus, although numerous studies have demonstrated that several factors are involved in the entrainment of the RPE circadian clock, further studies are needed to understand the mechanism controlling this phenomenon in the RPE fully.

**Figure 2 ijms-23-02699-f002:**
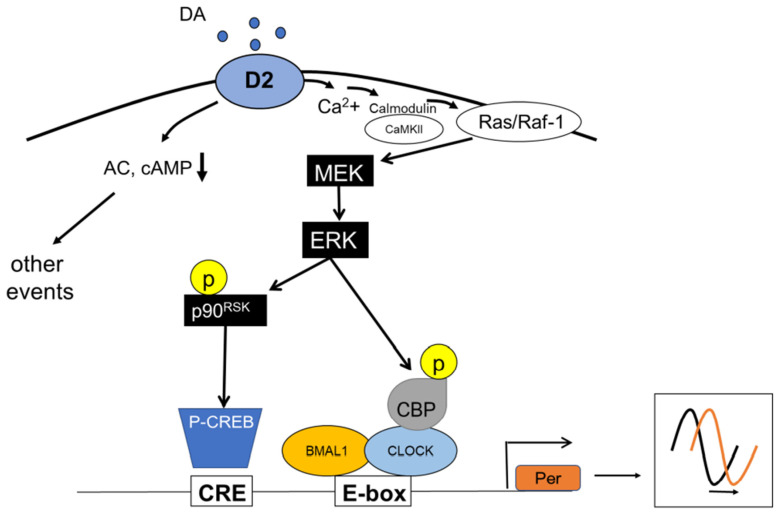
Dopamine clock-resetting pathway in the RPE. Schematic illustration of the proposed mechanism by which D_2_R signaling phase shifts the circadian clock in the RPE. Activating these receptors by dopamine (DA) leads to MEK activation, then Erk1/2 (ERK), and finally the phosphorylation of p90^RSK^, which in turn phosphorylates CREB at Ser133, thereby activating the *Per1/2* promoters. Alternatively, ERK can also induce *Per1/2* transcription via the phosphorylation of CBP, causing it to bind to and activate the BMAL1:CLOCK [90].

## 5. Is the Daily Burst in Phagocytic Activity Important for Photoreceptor and RPE Health?

As we have previously mentioned, several studies have supported the notion that the timing and presence of the daily peak in phagocytic activity play an important role in regulating the RPE and photoreceptor physiology and have speculated that a lack of this event may have a negative impact on these cells. Indeed, Nandrot et al. [29] reported that mice lacking ανβ5 integrin receptors fail to show the morning burst of phagocytic activity. During the aging process, these mice lose cone and rod photoreceptors more rapidly than wild-type mice, and the function of these cells is significantly impaired [29]. Thus, this study clearly indicated that the loss of the daily rhythm in RPE phagocytosis might reduce the photoreceptors’ viability during aging. A few years later, this group also reported that mice lacking MFG-E8 also do not show the phagocytic activity peak after light onset [30]. However, surprisingly, these mice did not show any significant difference during the aging process [90]. More recent investigations have also shown that mice lacking D_2_R or in which *Bmal1* has been removed from the RPE do not show any negative phenotype in the photoreceptor function and RPE morphology, even during aging [31,42].

Finally, a recent investigation reported that in germline MT_1_ KO mice, there is a significant reduction in the number of photoreceptors at 18 months of age [5] and an increase in lipofuscin-like (i.e., autofluorescence) accumulation in the RPE [34]. Initially, we attributed this effect to the fact that removal of MT_1_ affected the time of the phagocytic peak [34]. However, recent data suggest that the presence of the phagocytic peak does not affect RPE or photoreceptor health and viability during aging [31,42]. Hence, we now believe that the direct action of MT_1_ signaling on the RPE and/or photoreceptor cells are responsible for the phenotype(s) observed in our previous study.

Thus, experimental evidence from different mouse models suggests that the loss of the diurnal phagocytic peak of the POS does not result in any deleterious effects in the retina and RPE during aging. As mentioned earlier, importantly, in all the mice models described above, the RPE total daily phagocytic activity was not different, although the peak was abolished. Thus, the increase in the basal phagocytic activity may compensate for the peak loss.

## 6. Conclusions

In conclusion, experimental evidence accumulated over the last fifty years has demonstrated that the daily burst in phagocytic activity by the RPE is present in several vertebrate species (diurnal and nocturnal) as well as in the rod and cone photoreceptors. The mechanisms controlling this burst are complex and involve the circadian clock in the RPE and a well-known circadian output of the retina (i.e., dopamine) (Figure 3). However, new experimental evidence suggests that such a peak is not important for the health of the RPE or the photoreceptors.

## Figures and Tables

**Figure 1 ijms-23-02699-f001:**
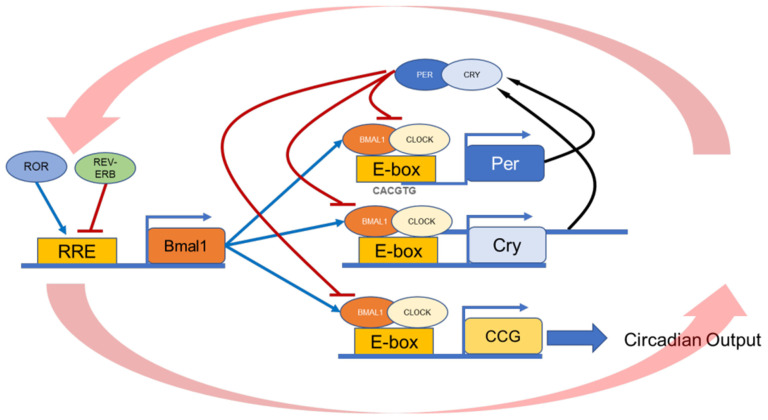
Schematic illustration of the molecular circadian clock. BMAL1:CLOCK heterodimer binds to the E-box present on the promoter region of the *Per* and *Cry* genes. Then, PERs, together with CRYs, inhibit their transcription by blocking the action of the BMAL1:CLOCK. The second feedback loop involves the transactivation of the *Rev-Erb* and *Ror* genes. REV-ERB and ROR compete for binding to RRE elements in the *Bmal1* promoter, driving a daily rhythm of *Bmal1* transcription. These feedback loops generate a 24-h rhythmic oscillation.

**Figure 3 ijms-23-02699-f003:**
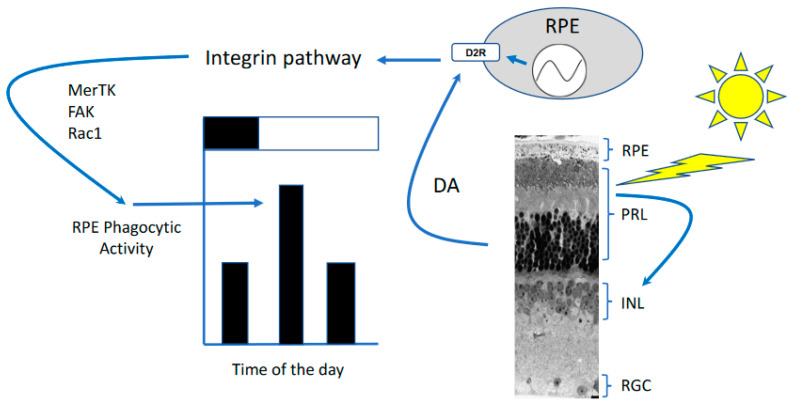
Proposed mechanisms by which light regulates the burst in phagocytic activity after the onset of light. Photoreceptors (PRL) perceive light and then signal to dopaminergic neurons in the inner retina (INL) to release dopamine (DA) [91]. DA then diffuses within the retina and reaches the RPE. DA binds to D_2_R and activate integrin signaling thus stimulating phagocytic activity by RPE after the onset of light. Thus, DA signaling, via the D_2_R, conveys the light signal to the RPE that regulates the burst in phagocytic activity. The circadian clock in the RPE also controls the burst in phagocytic activity by controlling the expression of D_2_R.

## Data Availability

Not applicable.

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
