# Peer review of "Circadian Regulation of Retinal Pigment Epithelium Function"

_ijms, 2022, doi:10.3390/ijms23052699_

Round 1

Reviewer 1 Report

Baba et al has comprehensively put together all the observations related to circadian clock and RPE function which is very useful for the field. However, this review could even be improved with the following suggestions.

  1. Please describe all the animal models used to study the circadian rhythms related to RPE function in a separate section which would benefit the readers who would like to use these models in future.
  2. Please mention the relevance of circadian rhythms to human RPE diseases if any described previously.
  3. I would add all the references regarding any sentence mentioned in this review as far as I know, as this is a review article. for example the following sentence needs more references. If you think you have only one study please mention the authors name when starting the sentence. Numerous studies have also shown that circadian clocks are present in several ocular tissues (e.g., retina, cornea, retinal pigment epithelium, etc.) where they control critical physiological functions [1]". 
  4. All the studies done previously on manipulating the circadian clock and study the function of RPE should be given more importance and should be described to the extent.
  5. What is the effect of circadian rhythm on previously know disease  mouse models with RPE loss of function is it known ?

Author Response

Thanks for your comments on our review. 

  1. We have added a table describing the current animal models and the relevance to human health 
  2. There is no study that has shown that circadian rhythms in humans affect the RPE. We have added a paragraph about this gap in knowledge
  3. The article we are quoting is a recent and comprehensive review on the topic. Since the review focus on the RPE we think that quoting a recent review is sufficient. However, the key reference about this topic are in the main text when text when we describe the specific functions
  4. There are only 2 studies that have investigate clock genes removal. We have rewritten a section to address this issue
  5. No study has addressed this topic.

Reviewer 2 Report

Thie review is interesting, comprehensive, and well written.  It provides an informative overview of circadian regulation in RPE of photoreceptor outer segment phagocytosis.  I thought figures 1-3 highlighted information in the text, and the corresponding figure legends clearly explained the presented data or illustrations.  However, figure 4 was difficult to understand and I couldn't decipher the model proposed by the authors regarding melatonin and dopamine receptors from examining this figure or the corresponding legend. The text within figure 4 was blurry and no explanation was provided for the right half of the figure. I suggest the authors modify figure 4 to more clearly present their model and to improve resolution.  Perhaps a timeline of events would work better than many arrows between the RPE and photoreceptors.

Author Response

Thanks for your review and suggestions.

We have redrawn Figure 4 and rewritten the caption as suggested